# Pegfilgrastim in Supportive Care of Hodgkin Lymphoma

**DOI:** 10.3390/cancers14174063

**Published:** 2022-08-23

**Authors:** Claudio Cerchione, Davide Nappi, Alessandra Romano, Giovanni Martinelli

**Affiliations:** 1Hematology Unit, IRCCS Istituto Romagnolo per lo Studio dei Tumori (IRST) “Dino Amadori”, 47014 Meldola, Italy; 2Department of Hematology and Cell Bone Marrow Transplantation (CBMT), Ospedale Centrale di Bolzano, 39100 Bolzano, Italy; 3Department of General Surgery and Medical-Surgical Specialties, Haematology Section, University of Catania, 95123 Catania, Italy; 4Division of Haematology, AOU Policlinico “Vittorio Emanuele”, 95123 Catania, Italy

**Keywords:** pegfilgrastim, Hodgkin lymphoma, supportive care, G-CSF, primary prophylaxis

## Abstract

**Simple Summary:**

Pegfilgrastim, the pegylated form of filgrastim (recombinant human GCSF) is widely adopted as supportive care for preventing neutropenia or febrile neutropenia episodes during chemotherapy. Neutropenia is directly cause of potentially severe infections and indirectly cause of treatment delivery delay. No guidelines address the pegfilgrastim role in the specific setting of Hodgkin lymphoma (HL). Since HL is a young-adult disease and shows mostly a very a favorable outcome after chemotherapy, treatment delay or dose reduction could potentially affect negatively the outcome. The aim of our review is to explore the current scientific literature on pegfilgratim use in HL, evaluating both observational than prospective trial. Moreover, analyzing the latter, we aim to define some practical suggestion about primary prophylaxis with pegfilgrastim in HL.

**Abstract:**

Neutropenia and febrile neutropenia are common and potentially life-threating events associated with chemotherapy treatment in Hodgkin lymphoma (HL). Neutropenia-related infectious events could be an issue both for direct clinical consequences and for delay in treatment delivery, affecting final outcomes in a potentially highly curable disease. Pegfilgrastim is the pegylated form of filgrastim, the recombinant form of human G-CSF, capable of prevent and mitigate neutropenic effects of chemotherapy, when adopted as primary prophylaxis in several hematological malignancies. No updated version of major international guidelines provides clear indication on prophylaxis use of pegfilgrastim in HL to prevent febrile neutropenia episodes in HL. Moreover, to date, scarce and non-uniform clinical experiences evaluating pegfilgrastim as prophylaxis in HL are present in the literature. Herein, we propose a brief summary of the literature data about efficacy and safety of the use of pegfilgrastim as primary prophylaxis in HL during chemotherapy treatment.

## 1. Introduction

Decreases in the absolute neutrophil count (ANC) (neutropenia) and neutropenic fever (NF) are common adverse events experienced by patients undergoing chemotherapeutic treatment, and are potentially serious and life-threatening. IDSA provided definitions of both neutropenia and neutropenic fever [1]. Neutropenia is defined as an ANC of 500 cells/mm^3^ or an ANC that is expected to decrease to <500 cells/mm^3^ during the next 48 h. Fever in neutropenic patients is defined as a single oral temperature of 38.3 °C (101 °F) or a temperature of 38.0 °C (100.4 °F) sustained over 1 h.

NF, despites advances in pharmacological prophylaxis and treatment, is still a relevant cause of morbidity and death in patients undergoing myelosuppressive chemotherapy, as well as a reason for delay in treatment delivery. Finally, clinicians should also take account of hospitalization and treatment cost of one or more NF episodes in a patient. Then, a prompt diagnosis and treatment intervention of the NF episode are mandatory, but a correct prophylaxis strategy is also essential to avoid a first or subsequent NF episode.

The risk of NF episodes over a myelotoxic chemotherapy treatment directly correlates with the duration and severity of neutropenia [2]. Defining the risk of each patient of developing a long and/or severe neutropenia before the chemotherapy treatment starts could be assessed either by considering non-patient-related factors (e.g., type of chemotherapy) or patient factor (e.g., disease features, infectious history, etc.). The updated EORTC guidelines on G-CSF prophylaxis define the risk category (intermediate or high, 10–20% or ≥20%) by the type of chemotherapy treatment adopted, drawing a complete list of regimens and their corresponding risk [3]. Aside from the myelotoxic treatment itself, several factors can contribute to the overall risk of longer and deeper neutropenia in a given patient. The internationally validated MASCC score system provides a tool aiming to identify the individual risk for a patient of a complicated course of neutropenia during the chemotherapy [4]. A score greater than 21 predicts a low (<10%) risk of complication if neutropenia occurs, with direct implication in clinical management of prophylaxis and possible treatment. ASCO guidelines on neutropenic patients do not clearly cite the G-CSF prophylaxis approach in this setting [5]. Both ESMO and AGIHO/DGHO provide exhaustive guidelines on proper diagnostic pathways, prophylaxis and treatment strategies of NF episodes in patient undergoing myelotoxic treatment [6,7]. Regarding the first line anti-infective treatment (antibiotics) of NF episodes, both scientific societies provide guidelines sorting patients by risk category: higher risk patients are to be treated as inpatients and lower risk patients as outpatients. The choice of antibiotic treatment lies outside the context of this review. Moreover, both ESMO and AGIHO/DGHO define G-CSF prophylaxis strategies in patients undergoing myelotoxic anti-cancer treatment, but no specific guidelines exist, to date, on NF prophylaxis in patients with lymphomas.

Hodgkin lymphoma (HL) is an uncommon B-cell malignancy, defined by the presence, in a tissue specimen (usually lymph node), of a few tumor cells together with plenty of microenvironment accompanying the cell population [8]. It is recognized that the incidence is higher in adolescents and young adult population [9]. Overall, not taking into account some differences in various histological subtypes (that is beyond the scope of this review), HL remains to date one of the most curable neoplastic diseases, with >80% of patients (aged < 60 years) able to be cured with a combination of chemo- and radiotherapy at diagnosis [10].

Therefore, a potentially highly curable disease such as HL should have minimal treated-related morbidity and mortality rates. The effort to reduce the incidence of NF, one of the most dangerous adverse events during the chemotherapy delivery span, should be a priority for clinicians.

Very few papers in the literature cite pegfilgrastim in HL and no collection of these scattered data is available, to date. Then, this lack in the medical literature and the assumption of a presumed efficacy of pegfilgrastim in HL, as it does in non-Hodgkin lymphomas (NHLs), are the major issues guiding this review.

## 2. Filgrastim and Pegfilgrastim Pharmacology

The major method to reduce NF incidence during chemotherapy (in all malignancies, not only in HL) is its prevention by the using human recombinant G-CSF (filgrastim), first introduced in 1991, a molecule with a boosting effect on granulopoiesis [11]. G-CSF support was found to be a valid supportive strategy in delivering dose-dense chemotherapeutic treatment in both solid and hematological neoplasms [12,13,14]. Filgrastim, in daily clinical practice, is usually administered as “on demand” multiple doses, when ANC drops below a secure level or a clinical condition mandates it (e.g., NF). This is because, when administered subcutaneously, filgrastim stimulates a fast ANC increase within 24 h, commonly requiring multiple daily injections to reach a sufficient ANC threshold (5 days, on average) [15]. The secondary prophylaxis strategy is widely adopted, but it bears some limitations inherent in its pharmacology, almost always requiring more the one administration. In fact, physiological filgrastim clearance is a complex model, briefly involving two major pathways. First, a neutrophil-related clearance, mediated by a surface receptor on mature neutrophils, acts by mimicking the physiological negative feedback of G-CSF production, as soon as ANC rises [16,17]. Secondly, a renally mediated clearance is also a relevant pathway leading to G-CSF excretion [18]. These two clearance pathways indeed play a role in the short half-life of filgrastim, leading to its inconvenient delivery modality of multiple injections. Pegylation is a chemical process which some biological and therapeutic molecules undergo whose result is to increase their molecular weight and structure, with relevant pharmacological implications [19]. Filgrastim pegylation acts mainly on minimizing the role of renal clearance, thus leading to an extended half-life, allowing a single administration over a treatment cycle [20]. This strategy of secondary prophylaxis with filgrastim is currently sharing its role with the primary prophylaxis strategy, adopting pegfilgrastim, a pegylated and long-acting form of human recombinant G-CSF. Indeed, the pharmacological profile of pegfilgrastim allows the same powerful effect on myeloid progenitors with the advantage of a single and fixed-dose injection given per cycle, thanks to its reduced renal clearance and extended half-life [21,22,23]. Simplifying the administration of pegfilgrastim as a single fixed dose was the major goal during its clinical development, but raised questions about its adequate exposure level in different-sized patients. Weight-based doses of pegfilgrastim in trials have shown a higher average concentration in overweight patients than in those with lower weight. Given that pegfilgrastim acts like protein whose molecular size ensures that it tends to distribute in the watery extracellular volume, it would be possible that overweight patients, with higher bodyweight but lower extracellular water, have a better exposure to pegfilgrastim than underweight patients [20]. On the contrary, underweight patients do not seem to be overexposed to pegfilgrastim when a single dose schedule is adopted and no major clinically relevant side effects are reported when compared to higher weight patients. Thus, as the standard clinical practice defines, the pegfilgrastim dose per cycle is 6 mg and it is only administered by subcutaneous injection [23]. These pharmacological features bring about notable advantages in terms manageability for clinicians and willingness to receive by patients.

## 3. What the Literature Reports on Pegfilgrastim in HL

Very few data exist in the literature about the specific setting of pegfilgrastim use as prophylaxis in HL treatment. Nevertheless, before evaluating the literature in an HL setting, there is some information that could be borrowed indirectly from some studies on the NHL setting or studies involving both HL and NHL patients. To date, only a few phase II or prospective studies define pegfilgrastim primary prophylaxis as at least comparable, if not superior, in terms of efficacy in preventing infectious events and avoidance of chemotherapy treatment discontinuation in non-Hodgkin lymphomas [24,25,26,27]. An open-label, randomized, phase II study had as its first endpoint the duration of grade 4 neutropenia in the first cycle in 50 elderly patients. The authors compared the outcome after 1:1:1:1 ratio randomization: pegfilgrastim 60 and 100 microg/kg on day 2, respectively, daily doses of filgrastim (5 microg/kg/day) from day 2, no growth factor support. Patients were treated with standard cyclophosphamide, doxorubicin, vincristine and prednisolone (CHOP) chemotherapy. When patient groups were imbalanced for risk factors such as marrow involvement and prior treatment, the duration of grade 4 neutropenia and the tolerability were more comparable for one pegfilgrastim administration than for daily filgrastim [24]. Another prospective study evaluated the incidence rate of neutropenic fever (NF)-related chemotherapy disruptions as the primary endpoint, comparing patients undergoing primary prophylaxis with pegfilgrastim to patients treated with filgrastim on demand. A total of 122 patients, equally divided into two cohorts by the type of G-CSF support, treated with bendamustine–rituximab for indolent NHL were analyzed. Pegfilgrastim was significantly associated with lower rates of NF-related chemotherapy disruptions (11.4% vs. 1.6%, in filgrastim and pegfilgrastim groups, respectively, *p* = 0.04) and reduced days of hospitalization due to NF episodes (18 vs. 6 median number of days, respectively, *p* = 0.04) [25]. A single-center, real-world retrospective study evaluated the role of G-CSF prophylaxis in 85 patients diagnosed with NHL and chronic lymphocytic leukemia, treated with bendamustine–rituximab. Stratifying patients into two groups, the G-CSF administered included both filgrastim or pegfilgrastim. In the first group (*n* = 47), G-CSF was administered for primary prophylaxis, while in the second (*n* = 38) G-CSF was adopted as secondary prophylaxis. The primary endpoint was the incidence of febrile neutropenia and grade 3 or 4 neutropenia, the secondary endpoint was incidence of fever, infection, hospitalization, dose delay or reductions and absolute neutrophil count. The two groups did not differ in primary and secondary endpoints, aside from a higher frequency of dose delays in secondary vs. primary prophylaxis patients (40% vs. 13%, respectively; *p* = 0.01) [26]. A retrospective, single-center study aimed to evaluate the efficacy of primary prophylaxis with filgrastim or pegfilgrastim to prevent NF in patient with aggressive B-cell lymphoma treated with an etoposide, prednisone, vincristine, cyclophosphamide, doxorubicin, rituximab (DA-R-EPOCH) schedule. In particular, the primary endpoint was to compare the median and average last dose level reached with DA-R-EPOCH in filgrastim and pegfilgrastim groups (N = 30 and 35, respectively). The secondary clinical outcome parameters evaluated were, among others, incidence of infections, NF, bone pain and hospitalization days. The pegfilgrastim group was, in terms of dose-intensity level of treatment (median and average last dose levels reached in patients who received at least four cycles) and in secondary outcomes, similar to the filgrastim group [27]. Fewer trials have compared the pegfilgrastim primary prophylaxis strategy to filgrastim prophylaxis, comprehending both NHL and HL patients. An open-label, randomized, phase II study had as the primary endpoint evaluation of the duration of grade 4 neutropenia after one cycle of toposide, methylprednisolone, cisplatin and cytarabine with the two different G-CSF strategies. The patient population was composed of relapsed or refractory Hodgkin or non-Hodgkin lymphoma cases receiving salvage treatment. The first group received single administration pegfilgrastim 100 μg/kg per chemotherapy cycle (*n* = 33), the second received daily subcutaneous injections of filgrastim 5 μg/kg (*n* = 33) until adequate ANC level. The mean duration of grade 4 neutropenia was comparable in both groups (2.8 and 2.4 days, respectively) and the incidence of grade 4 neutropenia was 69% and 68%, in pegfilgrastim and filgrastim groups, respectively [28]. Similarly, but more recently, a multicenter, randomized, double-blind phase III trial involved both NHL and HL patients to compare the efficacy and safety of pegfilgrastim with filgrastim. The first aim was to evaluate the duration of severe neutropenia after the first course of chemotherapy. The treatment schedule consisted in cyclophosphamide, cytarabine, etoposide and dexamethasone ± rituximab (CHASE(R)), administered in 3 consecutive days. The total 109 patients were randomized 1:1 to receive to either pegfilgrastim (*n* = 54) on day 4 of cycle 1 plus filgrastim placebo or daily filgrastim (*n* = 55) plus pegfilgrastim placebo. The mean duration of severe neutropenia was 4.5 and 4.7 days, respectively, in pegfilgrastim and filgrastim groups, highlighting no significant differences between the two prophylaxis strategies [29]. No phase III randomized clinical trials have specifically addressed the role of primary prophylaxis with pegfilgrastim in HL in both first line or relapsing disease settings or are going to be actively recruiting to date. Therefore, only fragmentary information on this topic can be summarized by other studies. A doxorubicin, bleomycin, vinblastine and dacarbazine (ABVD) regimen is adopted worldwide as the first line treatment approach in limited stage HL [30,31]. This chemotherapy regimen is administered every 14 days, with a dose-intensity profile that unavoidably may be affected by prolonged neutropenia or NF episodes. A prospective study on the safety and efficacy of pegfilgrastim as primary prophylaxis in ABVD delivery was performed on twenty-three patients diagnosed with classical HL at any disease stage [32]. Pegfilgrastim was given at a fixed dose of 6 mg once per cycle after 24 h of each ABVD dose delivery. Outcome measures evaluated were neutropenia, dose-delay, NF and dose-reduction incidence. Of all 230 ABVD doses delivered, only one was delayed because of neutropenia, no NF episodes were reported, ANC was reported above 500 in 99% of treatment cycles and no relevant side effects were reported. These results are consistent with a highly efficacious and safe profile as pegfilgrastim is administered as support for ABVD regimens. Additional data on pegfigrastim support along with ABVD administration could be extrapolated from a phase I trial addressed to find the MTD of doxorubicin in ABVD regimens [33]. Twenty-four subjects with advanced disease of classical HL (only one with nodular lymphocyte predominant HL) were administered with ABVD with three different doxorubicin dose levels in a dose-escalated fashion (35, 45 and 55 mg/m^2^, respectively), aiming to evaluate DLT and non-DLT to evaluate MTD. A primary prophylaxis with pegfilgrastim was administered after 25 h with each ABVD dose. Narrowing the field to ANC drop-related events, 11 episodes of infection were reported, but only one (grade 3 neutropenic infection) was of DLT significance, resulting in a 1-week delay of cycle delivery. This trial evidenced the possibility of delivering a higher dose of doxorubicin in ABVD regimens with pegfilgrastim support, giving a relatively safe profile of dose level of 35 and 45 mg/m^2^ compared to standard ABVD (neutropenia grade 3–4: 0, 42 and 34%, respectively) [34]. This trial demonstrated that pegfilgrastim primary prophylaxis is potentially also effective in protecting from neutropenic events in case of a higher dose-intensity ABVD regimen (higher doxorubicin doses, in this case). A more specific trial evaluated the role of pegfilgrastim in supporting full dose delivery of BEACOPP regimens, every 14 days as scheduled [35]. The superiority of BEACOPP regimens compared to ABVD in advanced stage HL was demonstrated as well as the feasibility and efficacy of a higher dose and dose density of BEACOPP administered every 14 days [36,37]. Forty-one high-risk HL patients were enrolled to receive BEACOPP-14 for up to eight cycles, as the primary objective was the proportion of patients receiving the chemotherapy at a full dose and on schedule (FDOS). Furthermore, patients were randomly (1:1) allocated to receive prophylactic pegfilgrastim at day 4 or 8 of each cycle. Overall, considering all planned cycles, 90% of these were delivered and 81% at FDOS, with neutropenia and thrombocytopenia as driven toxicity leading patients to not receive FDOS. Grade 3–4 neutropenia were at 67%, 85% and 76% for pegfilgrastim on day 4, day 8 and in total, respectively. A total of 11 episodes of NF (for eight patients) were reported, without affecting treatment delivery planning. These interesting results confirm the feasibility of delivering an intensive regimen such as BEACOPP-14 with the support of pegfilgrastim primary prophylaxis on days 4 or 8, but with an advantage in terms of neutropenia incidence for day 4 administration. Table 1 provides a summary of these reported data on pegfilgrastim.

## 4. Conclusions

There are no observational or prospective trials, to date, comparing filgrastim to pegfilgrastim as a prophylaxis strategy in HL. Indeed, as our previous experiences of treating other hematological neoplasms have shown, patients can benefit from a pegfilgrastim primary prophylaxis strategy to prevent neutropenia, NF episodes and treatment delay [38,39,40]. These experiences, in addition to the still scant data reported above, raise the question if primary prophylaxis with pegfilgrastim should be added as a standard option for supportive care of HL too. The advantage of delivering the full dose of treatment, without cycle delay, and the protection against potentially serious infective events, even if not common in the targeted population undergoing treatment for HL, is essential in a highly curable disease. With the limit of no clinical trial available, the choice of pegfilgrastim prophylaxis should be mainly based on the type of treatment delivered. EORTC guidelines cite the most adopted chemotherapy regimen in HL, both as first and second lines [3]. As ABVD bears a very low risk of neutropenia and, especially, infectious consequences, it should be considered whether filgrastim can be used as an on demand strategy or pegfilgrastim as a primary prophylaxis strategy. On the other side, a more aggressive schedule of treatment, such as BEACOPP, with a higher risk of neutropenia and infections, should be recommended with pegfilgrastim as supportive care. Finally, the intrinsic risk factors of the patients themselves (such as disease burden, age, comorbidity, etc.) can drive the choice, as the MASCC score suggests [4]. In our opinion, clinicians should consider pegfilgrastim as primary prophylaxis, instead of a filgrastim on demand strategy, in HL settings when aggressive treatments (e.g., BEACOPP) are planned. This approach, aiming to avoid NF episodes as much as possible, has the scope to deliver the full dose intensity of the treatment, that is crucial when a potentially curative approach is planned. On the other side, lower risk settings (e.g., ABVD) could be safely managed with both pegfilgrastim and filgrastim support, as clinicians’ choice. Otherwise, it is questionable if a regimen with low rates of severe or long-lasting neutropenia, as ABVD is, can benefit from anti-microbial prophylaxis in place of G-CSF support if neutropenia occurs. Clinicians must be aware of local policies on anti-microbial use and the growing issue of anti-microbial resistance. The economical cost/benefit ratio has also to be considered, given the higher cost of pegfilgrastim compared to filgrastim and the possible total cost of one or more NF episodes in a patient. In our opinion, in the high-risk context (e.g., BEACOPP), pegfilgrastim may also contribute to an overall cost reduction, avoiding NF episodes and potential subsequent treatment lines if first line treatment is not correctly delivered. As well as other hematological malignancies, in HL treatment there were notable advances with the entry of novel agents, which are even able to induce or potentiate neutropenia when added to classical chemotherapeutic agents. In the ECHELON-1 trial, brentuximab vedotin, an anti-CD30 conjugate, was compared to ABVD when added to AVD (A + AVD), showing higher rates of neutropenia, especially in patients aged > 60 years [41]. Interestingly, especially in the latter group of patients, a beneficial effect was observed in those receiving G-CSF primary prophylaxis (as per local policy), as shown by reduced rates of neutropenia and NF episodes (40% vs. 78% and 30% vs. 38%, respectively). Thus, in our opinion, is reasonable to extend the indication of a G-CSF primary prophylaxis even in the context of the newer agents already in use or in clinical trial settings. No practical disadvantages are seen with pegfilgrastim compared to filgrastim, in terms of side effects or toxicity. Conflicting data on a higher incidence of bone pain with pegfilgrastim are reported, nevertheless it has to be considered as a mild and easily manageable side effect [42,43].

## Figures and Tables

**Table 1 cancers-14-04063-t001:** Summary of reported data on pegfilgrastim.

References and Study Design	Patients Number and Disease Setting	Treatment	Pegfilgrastim Timing	Comments
Younes A. et al. [32]	- 23 - Early and advanced stage	ABVD+/−RT	24 h after each ABVD infusion	- No NF episodes reported - Overall, 1 dose-delay reported due to neutropenia
Gibb A. et al. [33]	- 24 - Advanced stage	ABVD with doxorubicin dose escalation	24 h after each ABVD infusion	- Favorable efficacy and safety profile in delivering higher doxorubicin doses
Engert A. et al. [35]	- 41 - Advanced stage	BEACOPP-14	Days 4 or 8 after each cycle	- Effective in delivery of treatment at full dose and on schedule

## Data Availability

The data can be shared up on request.

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
