# Peer review of "Pegfilgrastim in Supportive Care of Hodgkin Lymphoma"

_cancers, 2022, doi:10.3390/cancers14174063_

Round 1
Reviewer 1 Report
Cercguione et al. provide an interesting review of the use of pegfilgrastim as supportive treatment for Hodgkin lymphoma. Comments:
The review would be improved if the authors provided a better background around the whole controversy of the use of filgrastim and pegfilgrastim in HL. https://ashpublications.org/ashclinicalnews/news/1983/Is-pegfilgrastim-with-ABVD-in-Hodgkin-lymphoma
They also provide clear recommendations for when pegfilgrastim should be strongly considered for primary and secondary prophylaxis. The need to provide the reader with a clear opinion on when filgrastim should be considered and when pegfilgrastim could be considered. i.e. with BEACOPP or escalated BEACOPP. What about ABVD.
They need to comment on AVD+brentuximab vedotin where in the ECHELON-1 trial there was a high incidence of neutropenia and primary prophylaxis is recommended now – should pegfilgrastim be used?
Can they comment on the risk of increased bone pain when pegfilgrastim is used. Is there any data?
What do they mean by ‘narrow the field’ in line 122?
Author Response
Dear Reviewer 1,
Thank you for your comment and suggestions, and for your interest in our paper.
The review would be improved if the authors provided a better background around the whole controversy of the use of filgrastim and pegfilgrastim in HL. https://ashpublications.org/ashclinicalnews/news/1983/Is-pegfilgrastim-with-ABVD-in-Hodgkin-lymphoma: Page 6, lines 275-278, we provided a brief personal opinion on this comment
They also provide clear recommendations for when pegfilgrastim should be strongly considered for primary and secondary prophylaxis. The need to provide the reader with a clear opinion on when filgrastim should be considered and when pegfilgrastim could be considered. i.e. with BEACOPP or escalated BEACOPP. What about ABVD: Page 6, lines 269-283. Summarized authors opinion on pegfilgrastim vs. filgrastim was added.
They need to comment on AVD+brentuximab vedotin where in the ECHELON-1 trial there was a high incidence of neutropenia and primary prophylaxis is recommended now – should pegfilgrastim be used? Page 6, lines 283-293, a comment on neutropenia and gcsf prophylaxis in ECHELON-1 was added
Can they comment on the risk of increased bone pain when pegfilgrastim is used. Is there any data? Page 6, lines 293-296, bone pain is not a relevant side effect of pegfilgrastim and literature is scant, some notions were added
What do they mean by ‘narrow the field’ in line 122? It means “focusing on”
Reviewer 2 Report
Febrile neutropenia (FN) is one of the most serious adverse events in patients with hematological malignancies and chemotherapy. Pegfilgrastim is the pegylated form of filgrastim, the recombinant form of human G-CSF, capable of preventing and mitigating neutropenic effects of chemotherapy when adopted as primary prophylaxis in several hematological malignancies. No updated version of NCCN or ASCO guidelines provides a clear indication of prophylaxis use of pegfilgrastim in HL to prevent febrile neutropenic episodes in HL. In this review, the authors propose a brief summary of the literature data about the efficacy and safety of the use of pegfilgrastim as primary prophylaxis in HL during chemotherapy treatment.
Some minor points:
1. The authors should provide more information about febrile neutropenia.
2. What are the current methods for febrile neutropenia treatment?
3. Compare the good and bad of pegfilgrastim and other treatments for neutropenia and febrile neutropenia.
Author Response
Dear Reviewer 2,
Thank you for your comment and suggestions, and for your interest in our paper.
Febrile neutropenia (FN) is one of the most serious adverse events in patients with hematological malignancies and chemotherapy. Pegfilgrastim is the pegylated form of filgrastim, the recombinant form of human G-CSF, capable of preventing and mitigating neutropenic effects of chemotherapy when adopted as primary prophylaxis in several hematological malignancies. No updated version of NCCN or ASCO guidelines provides a clear indication of prophylaxis use of pegfilgrastim in HL to prevent febrile neutropenic episodes in HL. In this review, the authors propose a brief summary of the literature data about the efficacy and safety of the use of pegfilgrastim as primary prophylaxis in HL during chemotherapy treatment.
Some minor points:
- The authors should provide more information about febrile neutropenia.
Page 2, lines 50-74
- What are the current methods for febrile neutropenia treatment?
Page 2, lines 71-73. Only very essential information was added about this subject, because pharmacological treatment of FN is outside the scope of this review (GCSF prophylaxis),
- Compare the good and bad of pegfilgrastim and other treatments for neutropenia and febrile neutropenia.
Page 6, lines 269-283. A brief opinion of pegfilgrastim vs filgrastim was added. No other treatment were compared, because there are no other prophylaxis strategies other than GCSF (other intervention, as antibiotic therapy is not part of the review subject).
Round 2
Reviewer 1 Report
All issues are addressed
Author Response
Thank you